# Assessment of Skills of Caregivers Providing Care for Stroke Patients in East Kazakhstan Region

**DOI:** 10.3390/healthcare13010027

**Published:** 2024-12-26

**Authors:** Gulnaz K. Kairatova, Zaituna A. Khismetova, Dariga S. Smailova, Dinara S. Serikova-Esengeldina, Duman Berikuly, Kamila M. Akhmetova, Gulnar M. Shalgumbayeva

**Affiliations:** 1Department of Public Health, Semey Medical University, Semey City 071400, Kazakhstan; kairatovag@bk.ru (G.K.K.); zaituna.khismietova@mail.ru (Z.A.K.); dinara_esengeldina@mail.ru (D.S.S.-E.); dumik1983@mail.ru (D.B.); 2Scientific Department, Kazakh National Medical University named after S.D. Asfendiyarov, Almaty 050012, Kazakhstan; dari1904_90@mail.ru; 3Department of Public Health, Astana Medical University, Astana 010000, Kazakhstan; kamikhudaiberdina@gmail.com

**Keywords:** informal caregivers, continuum of care, guardian, rehabilitation, stroke survivors

## Abstract

Objectives: This study investigated stroke survivors and the characteristics of care management after discharge from hospital to home. The study aimed to identify caregiving difficulties and to assess mastery of skills in implementing recovery activities at home. This was a cross-sectional study. Methods: As part of the study, we interviewed 205 informal caregivers caring for stroke survivors. For descriptive statistics, frequencies, mean, medians, and standard deviations were calculated. Results: Caregivers determined that the main aim was to restore motor activity in 58% (*p* ≤ 0.021), and the difficulties they most often encounter during caregiving are difficulties related to obtaining information from medical personnel, or not understanding the information received in people under 50 years of age, which were indicated in 47.9% of cases, while in people over 50 years of age, this indicator was 49.5%. The emotional state of the informal caregivers is closely interrelated with the state of stroke survivors. The informal caregivers with a higher education are more resistant to these changes related to the patient’s health. Conclusions: The informal caregivers in East Kazakhstan face challenges such as inadequate preparation, limited resources, and emotional strain, hindering effective care. High rates of stroke-related impairments highlight the need for better rehabilitation programs and informal caregiver support. Future research should assess intervention programs and support systems.

## 1. Introduction

According to World Health Organization (WHO) estimates, 1.3 billion people suffer from serious disabilities. This is 16% of the world’s population, or every sixth person [1].

One of the main causes of disability is stroke. Stroke differs from many other conditions as it occurs suddenly when the blood supply to the brain is occluded or there is a bleed resulting in damage to surrounding brain tissue. It is responsible for a wider range of disabilities than any other condition [2].

The most recent findings from the Global Burden of Disease (GBD) 2019 study indicate that stroke remains the second leading cause of death globally and the third leading cause of combined death and disability, as measured by disability-adjusted life years (DALYs) [3].

According to the World Stroke Organization (WSO) 2022 global fact sheet, more than 12.2 million new strokes occur each year. Globally, one in four people over 25 years of age will suffer a stroke in their lifetime. There are currently more than 101 million people worldwide who have had a stroke. Six and a half million people die from stroke every year. More than 143 million years of healthy life are lost each year due to stroke-related death and disability [4].

The incidence of stroke in the Republic of Kazakhstan increased by 2.3 times from 189 in 2011 to 433.7 per 100,000 people in 2020. It can be noted that the dynamics of the mortality rate have relatively decreased. In addition, compared to 2011, in 2020 (78.49), there was a decrease of 15%. In 2011, a mortality rate of 92.36 per 100,000 people was registered [5].

In absolute numbers, more than 40,000 cases of strokes are registered annually in our country, of which only 5000 people die in the first 10 days and another 5000 during the first month after the stroke [6,7].

Increasing evidence suggests that exposure to ionizing radiation may be a potential risk factor for cerebrovascular diseases, including stroke [8,9]. The Semipalatinsk Nuclear Test Site (SNTS), situated in the East Kazakhstan region, served as a major testing ground for nuclear weapons which resulted in major environmental contamination in the region. Even though the SNTS was closed more than 30 years ago, the population of adjacent territories continues to face certain health problems such as cerebrovascular diseases. The investigation in [10] suggests that the exposed population develops stroke at a younger age and has worse outcomes in terms of survival. Therefore, stroke is a relevant issue for the East Kazakhstan region.

People who survive a stroke often face functional limitations, including difficulties with mobility, speech, cognition, and daily activities. These challenges can significantly impact their independence and quality of life, requiring targeted rehabilitation and ongoing support. Also, stroke survivors need early and long rehabilitation. Kazakhstan was part of the former Soviet Union. Most post-Soviet countries have inherited the Semashko model of healthcare system. According to this model, the healthcare system primarily focused on diagnosis and inpatient treatment. Rehabilitation and recovery were given almost no attention [11]. But currently, the government of Kazakhstan is implementing programs for the early rehabilitation of stroke survivors, including those recovering from stroke. However, this process is still in its early stages, and home care for stroke survivors remains a significant issue for Kazakhstan.

Stroke survivors and informal caregivers can exert reciprocal influence on one another throughout the caregiving process and within the context of their social interactions. A stroke survivor’s depression, anxiety, and stress symptoms decrease both the stroke survivor’s and the informal caregiver’s physical and mental quality of life [8,12]. The incidence of aphasia, cognitive deficits, oral apraxia, dysarthria, and dysphagia occurred, respectively, in 61.8%, 76%, 30%, 61%, and 39% of stroke patients during the acute phase. Dysphagia represents a significant complication among stroke survivors in the chronic post-stroke phase. The early detection of this condition using screening tools with high diagnostic accuracy is critically important [13,14]. Post-stroke dysphagia (a difficulty in swallowing after a stroke) is a common and expensive complication of acute stroke and is associated with increased mortality, morbidity, and institutionalization due in part to aspiration, pneumonia, and malnutrition [15].

Mental health diagnoses in the first 3 years after a stroke increase the risk of death by more than 10%. In spite of the fact that stroke survivors with post-stroke depression were younger and had fewer chronic illnesses than nondepressed patients, we found that the mortality risk was still higher in those with post-stroke depression and other mental health diagnoses. These data speak to the seriousness of post-stroke mental health diagnoses, since the presence of a mental health condition confers as much risk for subsequent mortality as many other cardiovascular risk factors [16].

Urinary incontinence can affect 40% to 60% of people admitted to hospital after a stroke, with 25% still having problems when discharged from hospital and 15% remaining incontinent after one year [17].

These limitations have a significant impact on quality of life and accompanying social participation in daily life, as well as having a physical and emotional impact on the lives of loved ones. Return to work (RTW) is a common goal for adults after stroke; however, post-stroke disabilities may limit occupational opportunities [18]. This problem is a major economic burden for the state, which means the broader societal and economic system, including governments or nations, which bear the financial and social costs associated with post-stroke disabilities. These limitations significantly affect the quality of life and social participation in daily activities while also imposing physical and emotional burdens on the lives of loved ones. The physical, psychological, social, financial, and economic costs associated with loss of productivity for stroke survivors are reported to be billions of dollars each year [19].

Table 1 illustrates comparative data on the size of payments to stroke survivors and the effects of morbidity in Kazakhstan and the United States.

The profound changes that can accompany stroke may create considerable uncertainty in caring for the affected person and may necessitate changes in lifestyle or behavior patterns of both caregivers and stroke survivors [21]. Care provided commonly includes the following: physical, such as assisting with personal hygiene, feeding or mobility, care and assistance with medication; emotional, providing company and listening to worries; or practical, helping with cooking, shopping, cleaning, etc. The amount of time spent caring varies from a few minutes to many hours a week [2].

Families are often required to provide essential care for their stroke survivors, which can, in many cases, become a burdensome task. Being an informal caregiver entails disruptions to family [22], social [23], and work [24] life, as this role increases an individual’s workload and exerts significant emotional strain, potentially jeopardizing their psychosocial health. The classic caregiver stress framework portrays an image of caregivers who sacrifice their own health to enable disabled relatives to continue to reside in the community. It is easy to imagine how this process could break down, with mentally and physically compromised informal caregivers eventually providing lower-quality care, perhaps leading over time to abuse or neglect and, ultimately, to negative health outcomes for the stroke survivors [25].

Informal caregivers report feelings of uncertainty, emotional distress, and the need for training and information [26]. They are more likely to have depressive symptoms, report fair to poor physical health, experience higher levels of strain, and are less likely to engage in health-promoting activities, such as regular exercise, balanced nutrition, preventive healthcare visits, and stress management, compared to non-caregivers [27].

If informal caregivers and families have poor arrangements, caregiving can have a negative impact on health and well-being. Informal caregivers often face significant challenges in managing the evolving needs of stroke survivors, which can lead to negative physical, emotional, and social consequences over time. Despite the growing recognition of the burden caregiving places on stroke survivors, there remains a gap in the literature regarding effective interventions to support informal caregivers. Current strategies often fail to provide informal caregivers with the tools and knowledge necessary to address their changing responsibilities and prevent burnout [28].

Emotional well-being plays a critical role in the overall health and effectiveness of informal caregivers. The emotional state of informal caregivers is often impacted by the physical, mental, and emotional demands of caregiving, which can lead to stress, burnout, and a decline in health. Understanding the emotional state of informal caregivers is therefore essential for identifying appropriate interventions and support systems [22].

Informal caregivers need assistance in learning strategies to address their changing needs and prevent the detrimental effects of caregiving over time [29,30].

Purpose of the study: to identify the emerging difficulties of stroke survivors after discharge from hospital to home and the most common stroke survivors’ health deficits encountered by informal caregivers.

## 2. Materials and Methods

### 2.1. Study Design

This cross-sectional study was provided in the East Kazakhstan region (East Kazakhstan Oblast and Abay Oblast) from January to March 2023. In order to study the opinions of informal caregivers caring for stroke survivors and to identify the difficulties encountered in the process of care at home, a questionnaire was developed. The questionnaire was developed independently, drawing on international research and experience, and underwent a validation process. It was adapted from the studies [31,32], then translated into Russian and Kazakh. To ensure the accuracy of the translation, the questionnaire was translated back into English from both Russian and Kazakh and compared with the original version. Validation was carried out through a pilot test involving 15 randomly selected individuals, who were interviewed to assess the reliability and appropriateness of the survey. The pilot testing results led to minor adjustments, and the final, revised version of the questionnaire was used in the present study. The questionnaire contains 24 questions. The questionnaire consists of a socio-demographic section and a main block of questions reflecting the peculiarities of care, difficulties and care skills of informal caregivers, and a question assessing the emotional state of the informal caregiver on a ten-point scale. To assess the emotional state of informal caregivers, a self-report measure was employed, using a ten-point scale. This scale allowed caregivers to rate their emotional state, with 1 indicating extremely poor emotional well-being and 10 indicating optimal emotional well-being. The structure of the questions was either dichotomous (i.e., yes/no answers) or evaluative on a scale from 1 to 10.

### 2.2. Study Participants

This study involved consecutively recruiting 205 informal caregivers of stroke survivors, who had been caring for stroke survivors for a minimum of one month after discharge from inpatient care. Of these, 121 were female caregivers and 84 were male caregivers. The average age of the respondents was 50 years. An informal caregiver was defined as an unpaid person who lives with a patient and/or is most closely involved in taking care of him/her at home and helps with the physical care of coping with the disease. The participants were recruited by a convenient sampling method using mass invitations of people to take a survey through social networks. Study participants were recruited by distributing a questionnaire via Google Forms. In addition to Google Forms, alternative data collection methods were used, such as paper surveys and phone interviews, to ensure a broader range of participants. To reach populations with limited access to the Internet, targeted outreach was conducted with community organizations or healthcare providers who helped distribute the survey in person or by phone. Also, informal caregivers were recruited from primary healthcare clinics where their stroke survivors were undergoing follow-up care after a stroke. The inclusion criteria were as follows: voluntary consent to participate in the study and informal caregivers of stroke survivors. The exclusion criteria were as follows: refusal to participate in the study. All study participants provided written consent after providing detailed information about the purpose of the study and confidentiality of personal data. Participants’ data were coded with a unique code. The correspondence between this code and the personal identification information was stored in a file that only the database custodian had access to. The others had access to the coded (secure) database.

Prior to data collection, the study received approval from the Semey Medical University Ethics Committee (Protocol №1, 22 October 2022).

### 2.3. Statistical Analysis

The obtained responses were stored in an automated Google spreadsheet and Excel file. For qualitative data, we used Pearson’s chi-square. A value of *p* < 0.05 was taken as statistically significant. Statistical analysis was performed using the SPSS version 20.0 program (IBM Ireland Product Distribution Limited, Ireland). Descriptive statistics such as frequencies (n), mean values (M), medians (Me), and standard deviations (SD) were calculated.

## 3. Results

We conducted an analytical study to describe the peculiarities of caring for a relative with stroke and possible difficulties in the process of care. The average length of time after stroke onset is 23.9 months (IQR = 13).

Table 2 provides an overview of the study population detailing its main characteristics. Of the total sample of 205 individuals, 121 (59.1%) identified themselves as female and 84 (40.9%) identified themselves as male. Most of the participants were between forty and sixty years old (M = 50.57; SD = 13.475). Regarding education level, 37.6% of the respondents had a tertiary education, 33.7% had a secondary education, 13.7% had an incomplete secondary education, 10.2% had a primary education, and 4.9% had a postgraduate education. The majority of participants indicated themselves as being unemployed (48; 23.4%), 38 indicated themselves as being civil servants (18.5%), and heavy physical laborers and professional athletes accounted for 1%.

Table 3 presents the answers to the questions concerning the availability of disability groups and restoration of work activity. In total, 143 respondents answered that stroke survivors did not restore their work activity after having a stroke. This is more than half of the respondents (69.7%), and only 62 (30.2%) of the participants answered that stroke survivors had returned to their previous jobs. To the question “Does your relative who had a stroke have a disability group?”, 80 participants (39%) answered positively. In total, 105 (51%) answered that they did not, and 20 people (10%) abstained from answering. Of the patients who received disability due to stroke, 15 people belong to group 1, 36 to group 2, and 35 to group 3.

Table 4 shows the percentage of stroke sequelae and the specifics of post-stroke care direction. Informal caregivers noted that the majority of post-stroke patients had disorders such as motor dysfunction (119).

Speech impairment as a consequence of stroke occurred in 39%. In 18.5% of stroke survivors, there were complications such as swallowing difficulties, sensory impairment in 44%, mental dysfunction in 38.5%, and urinary incontinence and stool disturbance in 11%.

In the answers to the question studying the specifics of care directions in the post-stroke period, it can be seen that the most frequent recovery efforts are those aimed at restoring coordination of movement (46.3%), restoring mental health (43.9%), and restoring motor activity (39.5). The least frequent recovery efforts were those aimed at restoring sleep (17.6%) and nutrition (12.2%) (Table 3).

In order to identify the existing difficulties of informal caregivers, we compiled questions to identify them.

Table 5 illustrates the difficulties experienced by informal caregivers following discharge home from hospital. The data presented in the table are divided into the age groups compared: those aged 49 years and under (46.8%) and those aged 50 years and over (53.1%). People 49 years and younger have the most difficulty in obtaining a referral for rehabilitation (59.4%) and communicating with the stroke survivors (45.8%) compared to people 50 years and older. Meanwhile, people 50 years and older have difficulties in obtaining information from medical personnel (49.5%) and obtaining medicines within the guaranteed scope of free medical care (56%) compared to the second comparable age group of 49 and younger. In total, 19.5% received stroke care allowance, and 80.5% did not. Younger informal caregivers (17–49 years) reported a higher percentage (59.4%) of difficulties in obtaining referrals for rehabilitation compared to older informal caregivers (50 years and older) at 56.9%. This indicates that younger informal caregivers might face more challenges in navigating the healthcare system for rehabilitation services. Older informal caregivers (50 years and older) reported a higher percentage (56.0%) of difficulties in obtaining medicines, compared to younger informal caregivers (50.0%). This suggests that older informal caregivers might encounter more barriers in securing necessary medications within the framework of guaranteed free medical care. A higher percentage of older informal caregivers (59.6%) reported communication difficulties compared to younger informal caregivers (54.2%). This could suggest that older informal caregivers may face greater challenges in establishing effective communication with stroke survivors, possibly due to the severity of post-stroke impairments or other factors like caregiving experience. These trends highlight that caregiving needs and challenges can vary by age, suggesting the importance of tailoring support systems to address the specific difficulties faced by informal caregivers of different age groups.

According to Table 6, it is more common for people to have tertiary education proficiency in psychological rehabilitation skills than pre-university education (35.6% vs. 22%). Similar responses can be seen for proficiency in occupational therapy skills (26% vs. 18.6%), kinesotherapy (18.4% vs. 13.6%), and physical therapy (21.8% vs. 16.9%), which is statistically significant (*p* = 0.000). In contrast, those with pre-university education have more skills in speech rehabilitation (18.6%) than those with tertiary education (14.9%).

Table 7 shows comparative data on post-stroke health effects depending on the period of recovery. The results of the survey showed that post-stroke complications affecting cognitive function within a year of stroke onset occurred in 46.7%; after 12 months, this index decreased by 3.7% and amounted to 43%. The same cannot be said for motor complications. Climbing stairs and moving around became more difficult after 12 months from the moment of stroke onset (54%), while this indicator before 12 months was 45.7%. In addition, complications in physiological needs, such as personal hygiene and dressing, increased by 3.6% (31.4% before 12 months vs. 35% after 12 months).

To the question “How do you feel about the multi-disciplinary trio?” (doctor–nurse–informal caregiver), 92.6% of respondents answered positively, and 2.9% had a negative attitude.

Figure 1 shows the emotional state of informal caregivers on a 10-point scale according to education level, where 1 means a “bad” emotional state, and 10 means “excellent”. The majority of individuals with primary (62%) and incomplete secondary education (57%) rated their emotional state between two and five points. People with secondary, higher, and postgraduate education turned out to be more emotionally stable. Their answer ranged from five to eight points.

## 4. Discussion

The purpose of this study was to determine the challenges stroke survivors face after discharge from hospital to home and the most common post-stroke health deficits experienced by informal caregivers. The study also aimed to assess the impact of the patient’s health on the informal caregiver’s emotional well-being. The majority of informal caregivers were female (59%, *p* = 0.01). The average age was 50 years (M = 50.57 (95% CI: 48.71–52.42) SD = 13.47).

Many stroke survivors are discharged home with functional limitations requiring assistance. Researchers at the University of Belgrade estimate that after three months, 70% of stroke survivors have reduced walking speed, and 20% remain confined to a wheelchair [33]. Paresis of the upper extremity is a frequent impairment following acute stroke, which occurs in up to three-quarters of stroke survivors [34].

Although partial improvement of motor dysfunction is experienced during the recovery phase, many stroke survivors experience long-term impairments affecting dexterity and motor control of the paretic upper extremity [35].

The results of our study also show that motor impairment is present in 58% of stroke survivors (*p* ≤ 0.021). Consequently, informal caregivers’ actions are most often aimed at rehabilitation activities to restore movement coordination (46.3%). In their scientific report, Sang Gu Ji and H C Chiu found that one of the most important goals of post-stroke rehabilitation is to restore the gait structure and achieve fast walking so that stroke survivors can perform their daily activities without complications [36,37].

The tasks of achieving this goal after discharge will fall on the shoulders of informal caregivers, who often feel unprepared. Moreover, in a Swiss study, many caregivers described feeling abandoned and lonely with no one to turn to after discharge [32]. According to our study, difficulties in obtaining information from healthcare staff or not understanding the information received occurred in 47.9% of those under 50 years of age, whereas in those over 50 years of age, this was 49.5%. These findings are supported by other studies and reports on the needs of stroke survivors and carers following discharge home [32]. This limitation of discretion is likely to be exacerbated by the lack of information that caregivers reportedly have about their rights to be assessed and supported, as well as the rights to support the person they are caring for. Just as commonly, communication difficulties are associated with post-stroke depression and with aphasia. Post-stroke depression occurs in approximately one-third of all ischemic stroke survivors [20]. Approximately 1/3 of stroke survivors will be diagnosed with aphasia [38]. Even mild forms of aphasia can negatively affect functional outcomes, mood, quality of life, social participation, and the ability to return to work [39]. Approximately 1/3 of stroke survivors will be diagnosed with aphasia. Even mild forms of aphasia can negatively affect functional outcomes, mood, quality of life, social participation, and the ability to return to work.

Effective transitional care from hospital to home can be achieved when health professionals take a partnership approach to individualized transitional care, developing the ability of stroke survivors and informal caregivers to orient and inform about health and social care services to meet their care needs.

According to a study by our domestic colleagues, the main causes of high morbidity and mortality rates from stroke in the regions of Kazakhstan are a shortage of physicians, inadequate primary healthcare, insufficient observation and treatment, and untimely hospitalization [40]. According to the results of our study, difficulties in obtaining a referral for rehabilitation measures occurred in 62.9% of cases, although proper and timely rehabilitation can help to reduce disability and mortality rates.

Previous studies of home-based rehabilitation programs for stroke survivors have shown that post-stroke care is complex and different from care for patients with other chronic diseases [32].

Recent healthcare policy discourses have accelerated efforts to minimize hospital stays and provide more care in the community and the patient’s home [41]. There are high expectations for individuals and family members to manage their rehabilitation and aftercare, including the coordination of services [42]. This highlights a real need to initiate high-quality self-management support earlier in the stroke pathway to enable individuals and families to manage the transition from hospital to home and live well after the experience of stroke [43].

The shortened hospital stay usually does not give family members enough time to learn what they need to know to take on the role of informal caregiver [29]. The majority of participants in this study had difficulty in caregiving due to lack of skills in speech rehabilitation (82.9%) and kinesotherapy management (84.4%). Poorly trained informal caregivers pose a threat to patient safety and may increase preventable re-hospitalizations [32].

Unpreparedness for caregiving, fear for the future, stress, anxiety, and having difficulties in home rehabilitation can negatively affect the emotional well-being of caregivers. The study assessed the impact of stroke survivors on informal caregiver emotional well-being using a 10-point scale. Participants with elementary and less than a high school education rated their emotional well-being as low (1 to 3 points), while those with a high school education and a college degree rated their emotional well-being between 4 and 7 points. Informal caregivers of stroke survivors reported feeling depressed, isolated, and lonely when they returned home [7,29]. Increased emotional distress on the part of caregivers, in turn, reduces the ability to deliver high-quality care to stroke survivors and negatively impacts stroke survivor outcomes [44,45].

The evaluation of changes in post-stroke outcomes according to the recovery period showed that stair climbing and mobility became more difficult after 12 months from the onset of stroke (54%), and the cognitive complication within one year from the onset of stroke was 46.7%, whereas this rate decreased by 43% after that time. In a study conducted in London, the rate of cognitive impairment within three months of stroke onset was 43.9% [45]. Prolonged physical and cognitive impairment are major barriers to reintegration into the workplace. As the results of our study show, only 30.2% of stroke survivors continued their work activities. A study by Adams et al. showed that the success rate of returning to work at follow-up was 48.9% [46]. Failure to reintegrate into society can reduce quality of life, health status, and mental health. This not only affects the emotional well-being and financial stability of both the individual and their family but has a financial impact on society as a whole through loss of productivity and dependence on publicly funded supplementary income [47].

It is known that recovery from stroke varies greatly. Post-stroke consequences and recovery period depend on many factors: firstly, on the type of cerebral blood flow disorder, as well as the scale and localization. Timely medical care, rehabilitation care measures, the initial state of health of the stroke survivors, age, comorbidities, etc., have not an insignificant role in recovery [48].

This study sheds light on the specific challenges faced by informal caregivers of stroke survivors in Kazakhstan, emphasizing cultural and systemic factors unique to the local healthcare context. The shortage of physicians, inadequate primary healthcare, and limited access to timely rehabilitation services in Kazakhstan exacerbate the burden on informal caregivers, who are often unprepared for their roles.

The study revealed that 62.9% of participants experienced difficulties in obtaining rehabilitation referrals, highlighting systemic barriers within Kazakhstan’s healthcare infrastructure. Additionally, informal caregivers’ lack of proficiency in essential rehabilitation skills, such as kinesotherapy (84.4%) and speech therapy (82.9%), emphasizes the need for targeted training programs to enhance caregiving quality and patient outcomes.

Emotional strain was a key finding, with many informal caregivers reporting feelings of isolation, depression, and anxiety, particularly those with lower educational levels. This strain is compounded by communication challenges with healthcare providers and the severity of the patient’s condition. These factors align with broader cultural expectations in Kazakhstan, where family members are often the primary informal caregivers.

The study also underscores the importance of individualized care plans that address both stroke survivors and caregiver needs. Tailored interventions, such as caregiver education and psychological support, are critical to improving transitions from hospital to home and reducing caregiver burden.

Policymakers should prioritize improving communication pathways between informal caregivers and healthcare providers, simplifying access to rehabilitation services, and integrating informal caregiver support into healthcare strategies to address these challenges effectively.

This cross-sectional study has some limitations that need to be considered when interpreting the results. In Kazakhstan, homecare of stroke survivors is an important issue. We did not find any information about caregiver’s knowledge, attitude, and barriers relating to care of stroke survivors in Kazakhstan. We attempted to contribute to this discussion. This study was conducted using a voluntary convenient sampling method, so the number of respondents who agreed to the survey was not so large, although we tried to cover the whole of the East Kazakhstan region. The study’s cross-sectional nature limits its ability to establish causal relationships. It only provides a snapshot of the challenges informal caregivers face at a specific point in time, without capturing changes over time. Factors such as the stroke survivor’s severity of impairments, comorbidities, or informal caregivers’ previous experience with healthcare could have influenced the results. These variables were not controlled for, which could impact the generalizability of the findings. This research was conducted within the framework of a doctoral dissertation. In the future, we plan to conduct a large study at the national level, which will help eliminate these limitations.

## 5. Conclusions

In the East Kazakhstan region, informal caregivers face significant challenges in providing home-based care, including inadequate preparation for rehabilitation activities and limited resources. These gaps contribute to emotional strain and informal caregiving difficulties, emphasizing the need for targeted training, psychological support, and better communication with healthcare providers. High rates of functional and cognitive impairments in stroke survivors highlight the importance of comprehensive rehabilitation programs involving caregivers. Addressing systemic issues, such as improving access to rehabilitation services and enhancing informal caregiver readiness, is crucial for better outcomes. Future research should explore the effectiveness of intervention programs in Kazakhstan, focusing on longitudinal studies to evaluate informal caregiver training, resource accessibility, and support systems.

## Figures and Tables

**Figure 1 healthcare-13-00027-f001:**
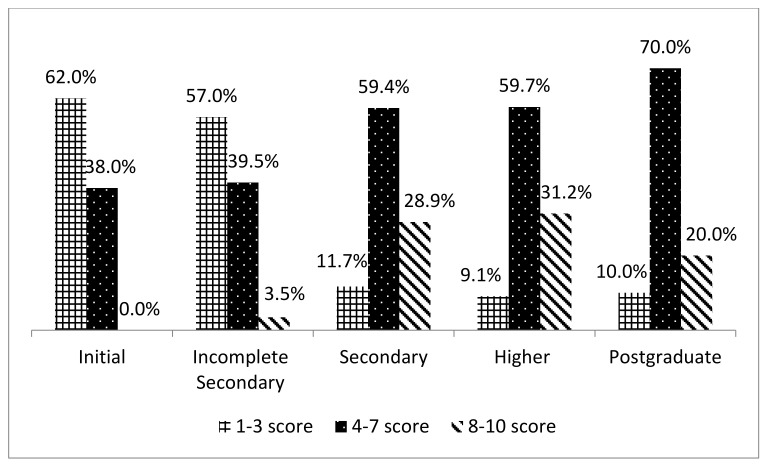
Assessment of the emotional state according to the level of education.

**Table 1 healthcare-13-00027-t001:** The volume of social disability benefits in the United States and Kazakhstan.

	Amount of Social Benefit	Subsistence Minimum
Kazakhstan	I group—KZT 95,496 *, USD 211.62 (as of 24 March 2024) II groups—KZT 76,397, USD 169.29 III groups—KZT 52,089, USD 115.43	From 1 January 2024, KZT 43,407 (USD 87.9) [20].
U.S.A.	A pivotal way those with compromised physical or mental conditions earn income is through Social Security Disability Insurance (SSDI) [16]. The vast majority of those who qualify for SSDI receive between USD 800 and USD 1800 each month.	The annual cost of living and hourly wages in cities across the United States range from USD 30,234 to USD 56,511 (2024) [12].

* All monetary values are expressed in tenge (KZT), the official currency of Kazakhstan.

**Table 2 healthcare-13-00027-t002:** Socio-demographic characteristics of informal caregivers.

General Characterization n = 205		n	%	*p*-Value
Gender	Males	84	40.9	*p* = 0.010
	Females	121	59.1	
Education	Initial	21	10.2	*p* = 0.000
	Lower secondary	28	13.7	
	Average	69	33.7	
	Higher	77	37.6	
	Postgraduate	10	4.9	
Occupation	Unemployed	48	23.4	*p* = 0.000
	Student	7	3.4	
	Housewife	20	9.8	
	Pensioner	28	13.7	
	Disabled person	3	1.5	
	State employee	38	18.5	
	Agricultural worker	7	3.4	
	Civil servant	34	16.6	
	Entrepreneur	12	5.9	
	Professional sportsman	2	1.0	
	Heavy physical labor worker	2	1.0	
	Worker in harmful and dangerous production	4	2.0	

*p*-values are based on statistical significance criteria, where values less than 0.05 (*p* < 0.05) indicate significant differences between the groups.

**Table 3 healthcare-13-00027-t003:** The quantity of stroke survivors who return to work and the presence of disability groups.

	Yes			No
Has your relative who suffered a stroke resumed his work?	30.20%			69.8%
Does your relative who has suffered a stroke have a disability group?	39%			61%
	I-group 7.3%	II-group 17.6%	III-group 17.1%	

**Table 4 healthcare-13-00027-t004:** Post-stroke consequences and recovery goals from the caregivers’ perspective.

The Aftermath of a Stroke			
**Questions**		**n (%)**	***p*-Value**
Does your relative have motor disorders, such as the consequences of a stroke?	No	86 (42)	χ2 = 5312 *p* ≤ 0.021
	Yes	119 (58)	
Does your relative have a speech disorder as a consequence of a stroke?	No	125 (61)	χ2 = 9878 *p* ≤ 0.002
	Yes	80 (39)	
Does your relative have a swallowing disorder, as a consequence of a stroke?	No	167 (81)	χ2 = 81,176 *p* ≤ 0.000
	Yes	38 (18.5)	
Does your relative have a sensitivity disorder, like the consequences of a stroke?	No	161 (78.5)	χ2 = 66,776 *p* ≤ 0.000
	Yes	44 (21.5)	
Does your relative have a violation of urine and stool excretion, as a consequence of a stroke?	No	182 (88.8)	χ2 = 123,322 *p* ≤ 0.000
	Yes	23 (11.2)	
Does your relative have a violation of mental functions (Neglect syndrome, Apathy, Agnosia), as a consequence of a stroke?	No	126 (61.5)	χ2 = 10,776 *p* ≤ 0.001
	Yes	79 (38.5)	
**Peculiarities of the direction of care in the post-stroke period**			
**Questions**	**n (%)**		***p*-value**
Restoring movement coordination	No	110 (53.7)	χ2 = 1098 *p* ≤ 0.295
	Yes	95 (46.3)	
Restoration of physical activity	No	124 (60.5)	χ2 = 9020 *p* ≤ 0.003
	Yes	81(39.5)	
Restoring power	No	180 (87.8)	χ2 = 117,195 *p* ≤ 0.000
	Yes	25 (12.2)	
Sleep recovery	No	169 (82.4)	χ2 = 86,288 *p* ≤ 0.000
	Yes	36 (17.6)	
Speech recovery	No	142 (69.3)	χ2 = 30,444 *p* ≤ 0.000
	Yes	63 (30.7)	
Restoration of disorders of mental functions	No	115 (56.1)	χ2 = 3049 *p* ≤ 0.081
	Yes	90 (43.9)	

*p*-values and χ2 values are based on statistical significance criteria, with *p* ≤ 0.05 and χ2 values indicating the strength of association between variables. *p*-values ≤ 0.05 are considered statistically significant.

**Table 5 healthcare-13-00027-t005:** Attitudes and barriers among caregivers during care of stroke survivors.

Questions		17–49 Years of Age		50 Years and Older		*p*-Value
		n	%	n	%	
Do you have any difficulties getting the necessary information from the medical staff for your relative who has suffered a stroke?	No	50	52.1	55	50.5	*p* ≤ 0.530
	Yes	46	47.9	54	49.5	
Do you have any difficulties getting a referral for rehabilitation activities for your relative?	No	39	40.6	47	43.1	*p* ≤ 0.000
	Yes	57	59.4	62	56.9	
Do you have any difficulties in obtaining medicines within the guaranteed scope of free medical care for your relative?	No	48	50.0	48	44.0	*p* ≤ 0.030
	Yes	48	50.0	61	56.0	
Do you have difficulty communicating with a relative who has suffered a stroke?	No	52	54.2	65	59.6	*p* ≤ 0.043
	Yes	44	45.8	44	40.4	

*p*-values are based on statistical significance criteria, where values less than 0.05 (*p* < 0.05) indicate significant differences between the groups.

**Table 6 healthcare-13-00027-t006:** Rehabilitation skills among informal caregivers.

Questions		Pre-Tertiary Education		Tertiary Education	
		n	%	n	%
The presence of skills in conducting speech rehabilitation	No	96	81.4	74	85.1
	Yes	22	18.6	13	14.9
Availability of skills in conducting psychological rehabilitation	No	92	78.0	56	64.4
	Yes	26	22.0	31	35.6
Having skills in conducting occupational therapy	No	96	81.4	64	73.6
	Yes	22	18.6	23	26.4
Having skills in conducting kinesitherapy	No	102	86.4	71	81.6
	Yes	16	13.6	16	18.4
Having skills in physical therapy	No	98	83.1	68	78.2
	Yes	20	16.9	19	21.8

**Table 7 healthcare-13-00027-t007:** Challenges for care of stroke survivors based on the recovery period.

The Most Difficult Care Activities	12 Months or Less		More Than 12 Months	
	n	%	n	%
Cognitive (speech, memory, reading, writing)	49	46.7	43	43
Physical activity (climbing stairs, moving around)	48	45.7	54	54
Physiological need (eating, personal toilet, dressing, bathing, etc.)	33	31.4	35	35

## Data Availability

The data presented in this study are available upon request from the corresponding author. The data are not publicly available due to privacy restrictions.

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
