# Peer review of "Assessment of Skills of Caregivers Providing Care for Stroke Patients in East Kazakhstan Region"

_healthcare, 2024, doi:10.3390/healthcare13010027_

Round 1

Reviewer 1 Report

Comments and Suggestions for Authors

The study addresses a critical issue—caregiving for stroke patients—which is both socially and medically significant.

Suggestions for Improvements:

Introduction:

It could be improved by explicitly linking this background to the study's aims. Clearly state how the study contributes to existing knowledge, especially its unique focus on East Kazakhstan. Provide a stronger connection between the global data on stroke and the specific challenges faced in Kazakhstan.

Methods:

Validation of the questionnaire: Include more details about how the questionnaire was validated for accuracy and reliability, especially after translation.

Sampling Bias: Address potential selection biases introduced by using Google Forms and suggest how this could be mitigated in future studies.

Include more details about the questionnaire development, sampling strategy, data handling, and validation processes to enhance transparency and replicability.

Data confidentiality: Expand on how participant data was securely managed and anonymized.

Results

Table Titles and Labels: The table titles could be more descriptive to ensure readers quickly grasp the content

Comparisons: While the results compare age groups and education levels, these comparisons could be emphasized more clearly to highlight trends.

Statistical Significance: While p-values are provided, discuss the clinical relevance of significant findings to highlight their practical implications.

Figure 1: The figure assessing caregivers' emotional states could be improved with clearer labels and a more descriptive caption to explain the findings.

Discussion:

Contextualization: Relate findings more explicitly to the local healthcare system and cultural factors in Kazakhstan.

Limitations: Acknowledge limitations, such as the cross-sectional design, self-reported data, and potential biases, and discuss their impact on the findings.

Conclusion:

Explicitly link conclusions to specific findings.

Include a discussion of study limitations.

Offer practical recommendations for healthcare providers or policymakers.

Future research: Specify areas for future investigation, such as longitudinal studies or intervention trials.

References:

Add Recent Studies: Ensure that the most recent studies (2022–2024) are included to reflect the latest trends in stroke care and caregiver challenges.

Author Response

Manuscript Revision

Journal:                      Healthcare

Manuscript No:          Healthcare-3366068

Manuscript title:        Assessment of Skills of Caregivers Providing Care for Stroke Patients in East Kazakhstan Region  

Authors:          Kairatova et al.

Response to Reviewer 1 Comments

Thank you very much for taking the time to review this manuscript. Please find the detailed responses below.

3. Point-by-point response to Comments and Suggestions for Authors

Comments 1: Introduction: It could be improved by explicitly linking this background to the study's aims. Clearly state how the study contributes to existing knowledge, especially its unique focus on East Kazakhstan. Provide a stronger connection between the global data on stroke and the specific challenges faced in Kazakhstan.

Response 1: Thank you for pointing this out. We agree with this comment. Therefore, I/we have added information p.2 line 54-71: “Increasing evidence suggests that exposure to ionizing radiation may be a potential risk factor for cerebrovascular diseases, including stroke. [8,9]. The Semipalatinsk Nuclear Test Site (SNTS), situated in the East Region of Kazakhstan, served as a major testing ground for nuclear weapons which resulted in major environmental contamination in the region. Even though the SNTS was closed more than 30 years ago, the population of adjacent territories continues to face certain health problems such as cerebrovascular diseases. The investigation [10] suggests that the exposed population develops stroke at a younger age and has worse outcomes in terms of survival. And therefore the stroke is relevant issue for East Kazakhstan region.

People who survive a stroke usually have to deal with various functional limitations and need in early and long rehabilitation. Kazakhstan was related to previous Soviet Union country. Most post-Soviet countries have inherited the Semashko model of healthcare system. According to this model the healthcare system primarily focused on diagnosis and inpatient treatment. Rehabilitation and recovery were given almost no attention [11]. But nowadays the government of Kazakhstan is implementing pro-grams for early rehabilitation of patients, including those recovering from stroke. However, this process is still in its early stages, and home care for patients remains a significant issue for Kazakhstan.

Comments 2:  Methods:

Validation of the questionnaire: Include more details about how the questionnaire was validated for accuracy and reliability, especially after translation.

Response 2: Agree. We have, accordingly, modified questionnaire validation to emphasize this point. p.4 line 156-165 “The questionnaire was developed independently, drawing on international research and experience and underwent a validation process. It was adapted from the studies (Lutz BJ,) and (Young ME), then translated into Russian and Kazakh. To ensure the accuracy of the translation, the questionnaire was translated back into English from both Russian and Kazakh and compared with the original version. Validation was carried out through a pilot test involving 15 randomly selected individuals, who were interviewed to assess the reliability and appropriateness of the survey. The pilot testing results led to minor adjustments, and the final, revised version of the questionnaire was used in the present study”

Comments 3: Include more details about the questionnaire development, sampling strategy, data handling, and validation processes to enhance transparency and replicability.  

Response 3: Agree. Done. We added information p.4 line 174-190: “This study involved consecutively recruited 205 informal caregivers of stroke survivors, who have been caring for stroke survivors for a minimum of one month after discharge from in patient care. Informal caregiver was defined as an unpaid person who lives with a patient and/or is most closely involved in taking care of him/her at home and helps with the physical care of coping with the disease. The participants were recruited by a convenient sampling method using mass invitations of people to take a survey through social networks. Also informal caregivers were recruited from primary healthcare clinics where their relatives were undergoing follow-up care after a stroke. Inclusion criteria were: voluntary consent to participate in the study, caregivers of stroke survivors. Exclusion criteria: refusal to participate in the study”

Comments 4. Data confidentiality: Expand on how participant data was securely managed and anonymized.

Response 4:  Agree. Done. p.4 line 190-194. We added information: “All study participants provided written consent after providing detailed information about the purpose of the study and confidentiality of personal data. Participants’ data were coded with a unique code. The correspondence between this code and the personal identification information was stored in a file that only the database custodian had access to. The others had access to the coded (secure) database”.

Comments 5. Results. Table Titles and Labels: The table titles could be more descriptive to ensure readers quickly grasp the content

Response 5: Agree. Done. We made the table titles more descriptive to ensure readers quickly grasp the content

Table 1 The volume of social disability benefits in the United States and Kazakhstan.

Table 2. Socio-demographic characteristics of informal caregivers

Table 3. The quantity of stroke survivors who return to work and the presence of disability groups

Table 4. Post-stroke consequences and recovery goals from the caregivers' perspective.

Table 5. Attitudes and barriers among caregivers during care of stroke survivors

Table 6. Rehabilitation skills among informal caregivers.

Comments 6. Comparisons: While the results compare age groups and education levels, these comparisons could be emphasized more clearly to highlight trends.

Response 6. Agree. Done. p.9, line 250- 263. We added information: “Younger caregivers (17-49 years) reported a higher percentage (59.4%) of difficulties in obtaining referrals for rehabilitation compared to older caregivers (50 years and older) at 56.9%. This indicates that younger caregivers might face more challenges in navigating the healthcare system for rehabilitation services. Older caregivers (50 years and older) reported a higher percentage (56.0%) of difficulties in obtaining medicines, compared to younger caregivers (50.0%). This suggests that older caregivers might encounter more barriers in securing necessary medications within the framework of guaranteed free medical care. A higher percentage of older caregivers (59.6%) reported communication difficulties compared to younger caregivers (54.2%). This could suggest that older caregivers may face greater challenges in establishing effective communication with stroke patients, possibly due to the severity of post-stroke impairments or other factors like caregiving experience. These trends highlight that caregiving needs and challenges can vary by age, suggesting the importance of tailoring support systems to address the specific difficulties faced by caregivers of different age groups”.

Comments 7. Figure 1: The figure assessing caregivers' emotional states could be improved with clearer labels and a more descriptive caption to explain the findings.

Response 7. Agree. Done. The figure assessing caregivers' emotional states was changed by improving clearer labels and a more descriptive caption to explain the findings.

Comments 8. Contextualization: Relate findings more explicitly to the local healthcare system and cultural factors in Kazakhstan.

Response 8. Agree. Done. p.11, line 388-409. We added information: “This study sheds light on the specific challenges faced by caregivers of stroke survivors in Kazakhstan, emphasizing cultural and systemic factors unique to the local healthcare context. The shortage of physicians, inadequate primary healthcare, and limited access to timely rehabilitation services in Kazakhstan exacerbate the burden on caregivers, who are often unprepared for their roles”.

The study revealed that 62.9% of participants experienced difficulties in obtaining rehabilitation referrals, highlighting systemic barriers within Kazakhstan’s healthcare infrastructure. Additionally, caregivers’ lack of proficiency in essential rehabilitation skills, such as kinesotherapy (84.4%) and speech therapy (82.9%), emphasizes the need for targeted training programs to enhance caregiving quality and patient outcomes.

Emotional strain was a key finding, with many caregivers reporting feelings of isolation, depression, and anxiety, particularly those with lower educational levels. This strain is compounded by communication challenges with healthcare providers and the severity of the patient’s condition. These factors align with broader cultural expectations in Kazakhstan, where family members are often the primary caregivers.

The study also underscores the importance of individualized care plans that ad-dress both patient and caregiver needs. Tailored interventions, such as caregiver education and psychological support, are critical to improving transitions from hospital to home and reducing caregiver burden.

Policymakers should prioritize improving communication pathways between caregivers and healthcare providers, simplifying access to rehabilitation services, and integrating caregiver support into healthcare strategies to address these challenges effectively.

Comments 9. Limitations: Acknowledge limitations, such as the cross-sectional design, self-reported data, and potential biases, and discuss their impact on the findings.

Response 9. Agree. Done. p.11, line 411-424. We added information: “This cross-sectional study has some limitations that need to be considered when interpreting the results. In Kazakhstan homecare of stroke survivors is actual issue. We didn’t find any information about caregiver’s knowledge, attitude and barriers relating to care of stroke survivors in Kazakhstan. We tried to clear this question. This study was conducted using a voluntary convenient sampling method, so the number of re-spondents who agreed to the survey was not so large, although we tried to cover the whole of East Kazakhstan region. The study's cross-sectional nature limits its ability to establish causal relationships. It only provides a snapshot of the challenges caregivers face at a specific point in time, without capturing changes over time. Factors such as the stroke survivor’s severity of impairments, comorbidities, or caregivers’ previous experience with healthcare could have influenced the results. These variables were not controlled for, which could impact the generalizability of the findings. This research was conducted within the framework of a doctoral dissertation, in the future we plan to conduct a large study at the national level, which will help eliminate these limitations”.

Comments 10. Conclusion:

Explicitly link conclusions to specific findings.

Include a discussion of study limitations.

Offer practical recommendations for healthcare providers or policymakers.

Future research: Specify areas for future investigation, such as longitudinal studies or intervention trials

Response 10. Agree. Done. p.11, line 426-236. We revised our conclusions to explicitly link them to specific findings and included a discussion of the study's limitations. Practical recommendations for healthcare professionals and policymakers were proposed, and areas for future research were identified. We added information: “In East Kazakhstan region caregivers face significant challenges in providing home-based care, including inadequate preparation for rehabilitation activities and limited resources. These gaps contribute to emotional strain and caregiving difficulties, emphasizing the need for targeted training, psychological support, and better communication with healthcare providers. High rates of functional and cognitive impairments in stroke survivors highlight the importance of comprehensive rehabilitation programs involving caregivers. Addressing systemic issues, such as improving access to rehabilitation services and enhancing caregiver readiness, is crucial for better outcomes. Future research should explore the effectiveness of intervention programs in Kazakhstan, focusing on longitudinal studies to evaluate caregiver training, resource accessibility, and support systems”.  

Comments 11. References:

Add Recent Studies: Ensure that the most recent studies (2022–2024) are included to reflect the latest trends in stroke care and caregiver challenges.

Response 11. Agree. Done. We updated the references where possible, incorporating recent studies. However, we had to retain those references whose exclusion would have altered the meaning of the article. p.13, line 472, p.13, line 474: p.13, line 483; p.13, line 489; p.13, line 491; p.13, line 474; p.13, line 496; p.14, line 501; p.14, line 503; p.14, line 505; p.14, line 508; p.14, line 527; p.14, line 531; p.15, line 549; p.15, line 551; p.15, line 553; p.15, line 556 

Reviewer 2 Report

Comments and Suggestions for Authors

Thank you for allowing me to review your work. Please see the attached. 

Author Response

Manuscript Revision

Journal:                      Healthcare

Manuscript No:          Healthcare-3366068

Manuscript title:        Assessment of Skills of Caregivers Providing Care for Stroke Patients in East Kazakhstan Region  

Authors:          Kairatova et al.

Response to Reviewer 2 Comments

Thank you very much for taking the time to review this manuscript. Please find the detailed responses below

3. Point-by-point response to Comments and Suggestions for Authors

Abstract

p.1 For the abstract and the rest of the paper, would suggest having the language to describe caregivers be consistent. You use the term relative, informal caregiver, relative carer, family and other terms throughout the paper. These can all mean different things. I recommend clearly defining who you looked at, so it is clear from the title, abstract and throughout.

Response 1: Agree. Based on your recommendation, we adopted consistent terminology in our article to describe caregivers. In the materials and methods section, we provided a clear definition of a caregiver as an individual caring for a sick relative. p 4. Line177-179.  Previously used terms such as "relative," "family caregiver," "family," and others were replaced with the unified term "informal caregiver."

Comments 2:  p. 1 As above I also recommend using the same language for the care recipient. If you are looking at people who have had strokes that are being cared for in the community, then they would be stroke survivors or care recipients or individuals who have suffered a stroke. Whatever term you decide on, I recommend using consistently throughout the paper.

Response 2: Agree. Based on your recommendation we adopted consistent terminology in our article to describe stroke survivors. We replace terms as care recipients or individuals who have suffered a stroke to the stroke survivor.

Comments 3: Introduction. p.1 line 34. This sentence reads a little awkwardly with “estimates” – is it possible to reword for clarity?

Response 3: Agree. p.1 line 36-39. We reword for clarity: “The most recent findings from the Global Burden of Disease (GBD) 2019 study indicate that stroke remains the second leading cause of death globally and the third leading cause of combined death and disability, as measured by disability-adjusted life years (DALYs)”.

Comments 4. p. 2 line 52. Here please identify the functional limitations before you go on to describe them in the subsequent sentences.

Response 4: Agree. p.2, line 63-67. We added information: “People who survive a stroke often face functional limitations, including difficulties with mobility, speech, cognition, and daily activities. These challenges can significantly impact their independence and quality of life, requiring targeted rehabilitation and ongoing support.”

Comments 5. p. 2 line 75. Here it is unclear what is meant by the “state” – do you mean your country, a particular population or the disease? Please clarify.

Response 5: Agree. p.3, line101-105. Me added information for clearing of meaning. The term "state" in this context refers to the broader societal and economic system, including governments or nations, which bear the financial and social costs associated with post-stroke disabilities. These limitations significantly affect the quality of life and social participation in daily activities while also imposing physical and emotional burdens on the lives of loved ones.  

Comments 6. p. 2 Table 1- I would recommend adding a footnote re: monetary unit tenge – as it is not described in text. This way if someone were to look at your table they would have a comparative understanding.

Response 6. Agree. p.3 Table 1. We added a footnote: “All monetary values are expressed in tenge (KZT), the official currency of Kazakhstan”

Comments 7. p. 3 line 96 – please clarify what type of activities are health promotion activities

Response 7. Agree. p.3, line 129-132. We rephrased this statement: They are more likely to have depressive symptoms, report fair to poor physical health, experience higher levels of strain, and are less likely to engage in health-promoting activities, such as regular exercise, balanced nutrition, preventive healthcare visits, and stress management, compared to non-caregivers”

Comments 8. p. 3 line 101- I think here you need another statement or 2 identifying what the issue is- is there a gap in the literature? Are there no interventions for caregivers? What are you trying to accomplish with this research? This way it will make it clear why there is a problem. Right now, it is just implied from your statements in the introduction, rather than implicit.

Response 8. Agree. p.3, line134-140. We added information: “Informal caregivers often face significant challenges in managing the evolving needs of stroke survivors, which can lead to negative physical, emotional, and social consequences over time. Despite the growing recognition of the burden caregiving places on stroke survivors, there remains a gap in the literature regarding effective interventions to support informal caregivers. Current strategies often fail to provide informal care-givers with the tools and knowledge necessary to address their changing responsibilities and prevent burnout [28].”

Comments 9. Methods p. 3 line 110 Please clarify how questionnaire items were validated? Were any questions part of an instrument? Please include any validity and reliability if so. Please also identify how you determined your questionnaire-were item variables selected from the literature? This is not clear in this section.

Response 9. Agree. p.4, line 156-165. We have, accordingly, modified questionnaire validation to emphasize this point. “The questionnaire was developed independently, drawing on international research and experience and underwent a validation process. It was adapted from the studies (Lutz BJ,) and (Young ME), then translated into Russian and Kazakh. To ensure the accuracy of the translation, the questionnaire was translated back into English from both Russian and Kazakh and compared with the original version. Validation was carried out through a pilot test involving 15 randomly selected individuals, who were interviewed to assess the reliability and appropriateness of the survey. The pilot testing results led to minor adjustments, and the final, revised version of the questionnaire was used in the present study”

Comments 10. p. 3 line 117 Please clarify how the survey was delivered so it is clear how anonymity was maintained-electronic?

Response 10. Agree. p.5, line 190-194. Done. We added information: “All study participants provided written consent after providing detailed information about the purpose of the study and confidentiality of personal data. Participants’ data were coded with a unique code. The correspondence between this code and the personal identification information was stored in a file that only the database custodian had access to. The others had access to the coded (secure) database”.

Comments 11. p. 3 line 119 Please clarify inclusion and exclusion criteria. How were participants recruited for the survey? Were there any biases in recruitment?

p. 4 line 134 Please add citations for p value threshold and for SPSS

Response 11. Agree. p.4, line 180-188. Done. Study participants were recruited by distributing a questionnaire via Google Forms. In addition to Google Forms, alternative data collection methods were used as paper surveys, phone interviews to ensure a broader range of participants. To reach populations with limited access to the Internet, targeted outreach was conducted with community organizations or health care providers who helped distribute the survey in person or by phone. Also informal caregivers were recruited from primary healthcare clinics where their stroke survivors were undergoing follow-up care after a stroke.

p.5, line 200-201. Statistical analysis was performed using the SPSS version 20.0 program (IBM Ireland Product Distribution Limited, Ireland).  

Comments 12. Results p. 4 line 150 Table 2 - Please add footnote to table with the p value criterion – so if someone goes right to table they understand

Response 12. Agree. We added a footnote: p-values are based on statistical significance criteria, where values less than 0.05 (p < 0.05) indicate significant differences between the groups.

Comments 13. p. 5 Table 4 – please add footnote to this table with p value criterion and X2 criterion – so if someone goes right to table they understand

Response 13. Agree. Done. p. 5, Table 2. We added a footnote:  p-values and χ2 values are based on statistical significance criteria, with p≤0.05 and χ2 values indicating the strength of association between variables. p-values≤0.05 are considered statistically significant

Comments 14. p. 6 line 184 – please put context to this statement- did you not expect more or less stroke survivors? The “only” 39% does not make sense—or was this an interpretation for your study? Please clarify

Response 14. Agree. After critically reviewing this section of the article, we decided to remove this sentence as it lacks logical coherence with the preceding statements and was our assumption.

Comments 15 p. 6 Table 5 – please add the p value to footnote as previous comments

Response 15. Agree. Done. p.5, Table 5. p-values are based on statistical significance criteria, where values less than 0.05 (p < 0.05) indicate significant differences between the groups.

Comments 17p. 7 line 205 – It is unclear how you determined emotional state. I wonder if in the introduction/background of the paper you can speak to the importance of this variable, so it is clear why this is important, and then clearly define what you are looking at in the methods section

Response 17. Agree. p.4, line 141-145. We added in the introduction/background of the paper statements: “The emotional well-being plays a critical role in the overall health and effectiveness of informal caregivers. The emotional state of informal caregivers is often impacted by the physical, mental, and emotional demands of caregiving, which can lead to stress, burnout, and a decline in health. Understanding the emotional state of informal caregivers is therefore essential for identifying appropriate interventions and support systems”

p.4, line 168-171. At in the methods section we added information: “To assess the emotional state of informal caregivers, a self-report measure was employed, using a ten-point scale. This scale allowed caregivers to rate their emotional state, with 1 indicating extremely poor emotional well-being and 10 indicating optimal emotional well-being”

Comments 18 p.8 Figure 1. I think this figure needs some type of explanation in the footnote describing what the changes are for the person who is just glancing at the figure before reading the article

Response 18. Agree. Done. The figure assessing caregivers' emotional states was changed by improving clearer labels and a more descriptive caption to explain the findings

Comments 19 Discussion p. 8 line 213 For much of this section you introduce information you had not previously discussed.

There is nice synthesis, but I would recommend not introducing new background information.

Rather have the information introduced in the introduction, then synthesize with your results.

Response 19. Agree. p.9, line 299-300. We rephrased this statement: “The purpose of this study was to determine the challenges stroke survivors face after discharge from hospital to home and the most common post-stroke health deficits experienced by informal caregivers. The study also aimed to assess the impact of the patient's health on the informal caregiver's emotional well-being”.